# ONESPIKE: ULTRA-LOW LATENCY SPIKING NEURAL NETWORKS

## ABSTRACT

With the development of deep learning models, there has been growing research interest in spiking neural networks (SNNs) due to their energy efficiency resulting from their multiplier-less nature. The existing methodologies for SNN development include the conversion of artificial neural networks (ANNs) into equivalent SNNs or the emulation of ANNs, with two crucial challenges yet remaining. The first challenge involves preserving the accuracy of the original ANN models during the conversion to SNNs. The second challenge is to run complex SNNs with lower latencies. To solve the problem of high latency while maintaining high accuracy, we proposed a parallel spike-generation (PSG) method to generate all the spikes in a single timestep, while achieving a better model performance than the standard Integrate-and-Fire model. Based on PSG, we propose OneSpike, a highly effective framework that helps to convert any rate-encoded convolutional SNN into one that uses only one timestep without accuracy loss. Our OneSpike model achieves a state-of-the-art (for SNN) accuracy of $81.92\%$ on the ImageNet dataset using just a single time step. To the best of our knowledge, this study is the first to explore converting multi-timestep SNNs into equivalent single-timestep ones, while maintaining accuracy. These results highlight the potential of our approach in addressing the key challenges in SNN research, paving the way for more efficient and accurate SNNs in practical applications.[1]

## 1 INTRODUCTION

*Spiking neural networks* (SNNs) have gained attention in the research community due to their potential for energy efficiency. They can often emulate the architectures of advanced *artificial neural networks* (ANNs). However, the non-differentiability and discontinuous nature of SNNs complicate the use of standard back-propagation methods typically found in ANN training. Prior work either directly trains SNNs (Fang et al., 2021; Zheng et al., 2021; Datta et al., 2022) or transfers weights from trained surrogate ANNs to SNNs (Zhang et al., 2022; Bu et al., 2023; Ho & Chang, 2021). However, these existing works either suffer from lower accuracies or require functionalities not found in SNNs that often erode their energy competitiveness. It is therefore critical to align with the essential characteristics and prerequisites of SNNs to enhance compatibility with neuromorphic hardware.

The energy efficiency advantage of SNNs is based on their ability to achieve accuracies that are competitive with state-of-the-art ANNs as well as low latencies. Higher latencies translate directly to longer inference time and hence higher energy cost. In *rate-encoded* SNNs(Rueckauer et al., 2016), latency is equivalent to the *time window size*, denoted as $T$, which could be a constant for a given model. Besides energy, the value of $T$ is also crucial for real-time applications(Dethier et al., 2013; Pearson et al., 2007). Thus, when developing SNNs, the two-fold challenge lies in improving accuracy and reducing $T$. Recent methods that achieved over 70% accuracy on the Imagenet dataset(Deng et al., 2009) and their associated time window sizes are presented in Table 1. In direct SNN training, the current best time window configuration is $T = 4$ for achieving comparable performance. In the case of ANN-to-SNN conversion methods, larger time window sizes are required.

Existing methods have limitations, one of which is relying on $T > 1$, often causing significant accuracy trade-offs. None, for example, has surpassed the 80% accuracy mark on the ImageNet

---

[1]Code available for review at: https://anonymous.4open.science/r/OneSpike

Table 1: Time window sizes ($T$) of the state-of-the-art SNN methods and the accuracies they achieved on the ImageNet dataset.

| Method | $T$ | Accuracy |
|---|---|---|
| QCFS-SNN (Bu et al., 2023) | 64 | 72.85 |
| TCL-SNN (Ho & Chang, 2021) | 35 | 70.75 |
| Spikformer (Zhou et al., 2022) | 4 | 74.81 |
| Spikingformer (Zhou et al., 2023) | 4 | 75.85 |

dataset. To address this, we introduce a novel approach for converting any SNN with $T > 1$ into an equivalent one with $T = 1$ in such a way that accuracy is not compromised. In fact, in some cases, it even enhances the model's performance. Our method, *OneSpike*, incorporates a novel method we call *parallel spike generation* (PSG). In the OneSpike model, all network layer operations can be executed in a single timestep, resulting in significantly reduced latency. Furthermore, PSG produces all the spikes in a single timestep in parallel, effectively harnessing global information across spikes. This results in enhanced accuracy compared to conventional *integrated-and-fire* (IF) models in some cases. Notably, we achieved a top-1 accuracy of 81.92% on the ImageNet dataset using a OneSpike model with $T = 1$. This may bring to mind *binary neural networks* (BNN). However, BNNs operate quite differently from SNNs. More importantly, as we will show in Section 6.1, the accuracies of state-of-the-art BNNs are significantly lower than that of OneSpike.

In addition, we will also show the hardware feasibility of OneSpike, and explore the impact of weight quantization. In particular, we will show that OneSpike shows good robustness and retains a high accuracy even after weight quantization, enhancing its potential for further energy saving.

The contributions of this work can be summarized as follows:

1. We propose the *parallel spike generation* (PSG) method that generates all spikes for a network layer within a single timestep. Models incorporating PSG exhibit superior performance compared to those using the original IF model, especially when the time window size reduces. Meanwhile, we thoroughly discuss the feasibility of hardware implementation for the PSG method.

2. We introduce OneSpike, a framework that converts SNNs with $T > 1$ into SNNs with $T = 1$. Utilizing multiple-timestep SNN models from the CIFAR-10 dataset(Krizhevsky et al., 2009) as our reference points, we transitioned them into OneSpike configurations. This conversion resulted in a marked accuracy surge for low-timestep models, surpassing the original SNNs underpinned by the IF model. To the best of our knowledge, this is the first work on converting an SNN to a lower latency SNN.

3. We present a series of high-accuracy, ultra-low-latency SNNs based on PSG as OneSpike models. We achieved 81.92% SNN accuracy on the complex ImageNet dataset within one timestep. To the best of our knowledge, this is the first SNN model achieving 80% on the ImageNet dataset.

The paper is organized as follows: Sections 2 and 3 introduce related works and SNNs, including ANN pre-training and existing ANN-to-SNN conversion methods. In Section 4, we introduce our Parallel Spike Generation(PSG) algorithms and present the OneSpike framework. Section 5 presents the performance of our methods on the ImageNet dataset. Section 6 discusses binary neural networks (BNNs) and weight quantization to further reduce the model size, respectively. This is followed by a conclusion.

## 2 RELATED WORKS

### 2.1 METHODS OF TRAINING SNNS

The two most common ways of developing SNN models are direct training and transferring from ANNs to SNNs. In this paper, we shall focus on the more popular form, i.e. rate-encoded SNNs.

In direct training, techniques such as *backpropagation through time* (BPTT) and *surrogate gradient* (SG) are employed to handle the temporal properties and non-differentiability of SNNs (Fang et al., 2021; Zheng et al., 2021; Datta et al., 2022; Zhou et al., 2023; Li et al., 2021b). This approach can achieve lower latency, often using only a few time steps for complex tasks. However, direct training faces challenges during the training phase, requiring significant computational resources as well as issues with accurate gradient approximations. Moreover, direct training methods have not yet achieved competitive accuracy compared to state-of-the-art ANNs.

ANN-to-SNN conversion methods (Hu et al., 2021; Li et al., 2021a; Ho & Chang, 2021; Bu et al., 2023; Panda et al., 2020) focus on transferring the weights of an ANN model to an SNN model without further training. The key is to estimate the spiking rate in an SNN by activations of an ANN. Previously proposed approaches adopted techniques like weight normalization (Diehl et al., 2015) and temporal switch coding (Han & Roy, 2020) to achieve higher accuracy. An existing ANN can be easily converted to an SNN as long as it satisfies certain constraints. Given the abundance of tools and packages available for building ANNs, this method facilitates the acquisition of SNNs without requiring meticulous training. The shortcoming of existing ANN-to-SNN conversion methods is that they require large time window sizes. Thus, this paper aims to investigate minimizing time steps as well as high accuracy.

## 2.2 BINARY NEURAL NETWORKS (BNNs)

*Binary neural networks* (BNNs) (Courbariaux et al., 2016) constraining the weights and/or activations of neural network models to binary values (most using $+1/-1$). They have their own training algorithms based on binarization. Recent studies (Liu et al., 2018; Bethge et al., 2019; Tu et al., 2022) have also explored BNNs that are inspired by established architectures like ResNet and DenseNet by incorporating shortcut connections to enhance performance.

SNN models with a time step of one outwardly resemble BNNs. However, BNNs and SNNs differ in several aspects. BNNs use normal activation functions, quantized parameters, and traditional forward and backward propagation algorithms, much like standard ANNs. In a way, they can be viewed as extremely quantized ANNs. In contrast, SNNs utilize spike-based encoding and time-dependent computations. The integrate-and-fire rule is unique to SNNs with no parallel in BNNs. Both aim at energy efficiency and computational speed, making them suitable for low-power and high-efficiency applications. Unlike BNNs, SNNs also do not use softmax, swish, or any complex functions. While the use of softmax (Liu et al., 2022) or swish (Darabi et al., 2018) functions in BNNs improves accuracy, they also lead to increased computational overhead and pose challenges for hardware implementation commonly required for embedded settings.

## 3 PRELIMINARY

This section elucidates the foundational Integrate-and-Fire (IF) model utilized by SNN neurons for spike generation and our adopted techniques for converting ANNs into SNNs.

### 3.1 INTEGRATE-AND-FIRE MODEL

The IF model is the most popular SNN model (Bu et al., 2023). It offers a simple representation of how neurons accumulate membrane potential and fire spikes. In the IF model, the membrane potential $V$ of a neuron is treated as a capacitor that accumulates the influence of input currents over time. It is described by the following differential equation:

$$\tau_m \frac{dV}{dt} = I_{\text{syn}}(t) - V(t) + V_{\text{rest}} \tag{1}$$

Here, $\tau_m$ represents the membrane time constant, $I_{\text{syn}}(t)$ denotes the synaptic input current, and $V_{\text{rest}}$ signifies the resting potential. When the membrane potential $V$ crosses a certain threshold $\theta$, the neuron generates an action potential(spike). In an SNN, for layer $l$, the output spike $s_l$ at timestep $t \in T$ can be calculated as:

$$s_l(t) = \begin{cases} 1, & \text{if } V_l(t) \geq \theta \\ 0, & \text{else} \end{cases} \tag{2}$$

where $\theta$ is the firing threshold and the membrane potential $V_l(t)$ can be denoted as:

$$V_l(t) = V_l(t-1) + W_l s_{l-1}(t) - \theta \cdot s_l(t) \tag{3}$$

where $W_l$ is the weight, $s_{l-1}$ is the spike input from the previous layer and $V_l$ is initialized as 0.

## 3.2 ANN-TO-SNN CONVERSION

To convert ANNs to SNNs in a near lossless way, we used the *value-range* (VR) *encoding*(Yan et al., 2022) and *clamp and quantize* (CQ) (Yan et al., 2021) training. Specifically, we will clamp and quantize the activation to a discrete set of values, $\{T_{\min}/T_q, T_{\min}+1/T_q, T_{\min}+2/T_q, ..., T_{\max}/T_q\}$, mimicking spike trains in SNNs. In this paper, we set $T_{\min}$ to 0 for all cases as 0 holds significant prominence in the distribution. We define $T_c$ as the clamp level and $T_q$ as the quantization level, such that:

$$T_c = \frac{T_{\max}}{T_q} \tag{4}$$

Consequently, all activations are clamped to the range $[0, T_c]$, and the time window size becomes $T = T_q$. The clamp operation also simulates the behavior of the ReLU function. The weights and biases of the SNN model denoted as $\hat{W}$ and $\hat{b}$ respectively, are related to those of the equivalent CNN model, represented as $W$ and $b$. The original $l^{th}$ layer of the ANN with ReLU activation function, can be represented as:

$$x_{l+1} = \text{ReLU}(W_l x_l + b_l) \tag{5}$$

After the weight and bias conversion, the $l^{th}$ layer of the SNN would be:

$$x_{l+1} = \mathcal{S}(\hat{W}_l x_l + \hat{b}_l) = \mathcal{S}(\frac{T_c}{T_q} W_l x_l + b_l) \tag{6}$$

where $\mathcal{S}$ refers to the spiking neuron that generates the spike. The specific clamp and quantization levels for each model can be found in the appendix.

## 4 METHODOLOGY

In this section, we present the overview of the OneSpike model as well as the *parallel spike generation* (PSG) method. Our approach transforms any SNN with convolutional and fully connected layers that require $T > 1$, into an SNN with $T = 1$.

### 4.1 OVERVIEW OF ONESPIKE

Figure 1 illustrates a spiking convolutional layer within OneSpike. Fully connected layers are implemented in a similar way.

In converting the $T > 1$ SNN to OneSpike, we reuse channels as depicted in Figure 1. The channels are grouped, with each *group* consisting of channels associated with distinct kernels, while the weights are shared across the groups. In the case of a fully connected layer, the different groups would each consist of a fully connected layer.

Each group is responsible for processing a specific subset of feature representations. The $k^{th}$ group processes the spikes generated at $t = k, t \in T$ in the original SNN. As each layer receives a set of spike trains ($T = 1$) as input, we directly map spike trains from different groups to channels within their respective groups for computation. This eliminates the need for slicing and simplifies the allocation of subsets, streamlining the process.

For the original spiking convolutional layer $l$ with an input size of $[H, W, C, T]$ where $H$ and $W$ represent the height and width of the feature maps respectively, and $C$ represents the number of channels, while $T$ denotes the time window size, the procedure is summarized as: The number of channel groups denoted as $g$, is the original time window size $T$. For layer $l$ with a channel group $g_l$,

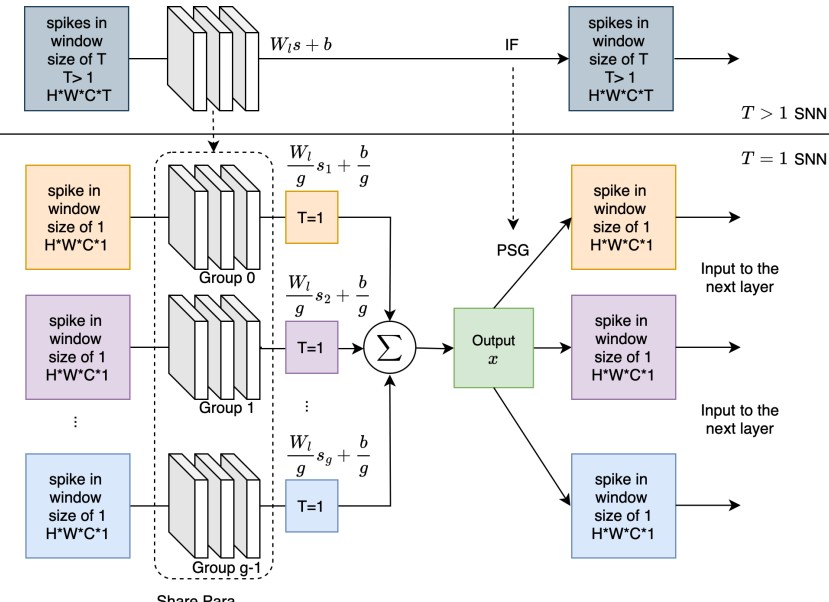

Figure 1: A spiking convolutional layer of OneSpike.

the input $s^{l-1}$ consists of $g_l$ groups of spike trains, each with a length of 1 and a size of $[H, W, C]$. The input for the $i^{th}$ channel group is represented by $s_i^{l-1}$. Within layer $l$, the output feature map $x_i^l$ for each channel group $i$ is computed as:

$$x_i^l = \sum_{j=0}^{n} W_j^l \cdot s_i^{l-1} + b^l \tag{7}$$

where $n$ is the number of channels in this group. Subsequently, we obtain the layer's output $s^l$ by taking the average of the feature maps as the output of layer $l$:

$$x^l = \sum_{i=0}^{g_l} x_i^l / g_l \tag{8}$$

Now, $g_l$ can be merged into the weight of the previous convolutional layer, eliminating the need for division operations as well as enhancing the feasibility of implementing our approach on hardware. So we have:

$$x^l = \sum_{i=0}^{g_l} \sum_{j=0}^{n} \frac{W_j^l}{g_l} \cdot s_i^{l-1} + \frac{b^l}{g_l} \tag{9}$$

$s^{l-1}$ is a spike train composed of 0s and 1s. As $T = 1$ in our case, $s^{l-1}$ is either 0 or 1. The multiplication of weights with $s^{l-1}$ can be efficiently implemented as addition operations in hardware, leading to reduced energy consumption.

## 4.2 PARALLEL SPIKE GENERATION (PSG)

We now introduce the PSG method to produce spikes from $x^l$ that serve as input for the subsequent layer, $l + 1$. Let $g_{l+1}$ be the count of channel groups in layer $l + 1$. From this, $g_{l+1}$ spike trains of dimensions $[H, W, C, 1]$ are concurrently generated. For the $i^{th}$ group in layer $l$, the membrane potential can be calculated as:

$$V_i^l = ((i - 1)x^{l-1} \bmod \theta) + x^{l-1} \text{ where } \theta \in \{2^n \mid n \in \mathbb{N}\} \tag{10}$$

The input for group $i$ is the $i^{th}$ spike in the train generated by the original SNN model. With the IF model, the previous $i - 1$ input would be accumulated through time. Once the threshold is

met, a spike is produced, and the threshold is subtracted from the membrane potential. Given our consistent input across timesteps, the residual membrane potential after $i-1$ steps is represented by $(i-1)x^l \bmod \theta$.

Then, the spikes generated by PSG would be:

$$s_i^l = \begin{cases} 1, & \text{if } ((i-1)x^{l-1} \bmod \theta) + x^{l-1} \geq \theta \\ 0, & \text{else} \end{cases} \quad \text{where } \theta \in \{2^n \mid n \in \mathbb{N}\} \qquad (11)$$

According to equation 11, given the previous layer's output and group index, all spikes can be generated independently within their respective groups. This facilitates the calculation of the corresponding membrane potential and the subsequent spike generation. To understand this better, consider an example where the output of layer $l$ has an element $x^l = 1.5$, $g_{l+1} = 4$, and a spike threshold, $\theta = 2$. Consider $i = 3$, $V_3^l$ in this case can be computed as follows: $V_3^l = ((1.5 * 2) \bmod 2) + 1.5$. All the values of $V_i^l$ are computed as $[1.5, 3, 2.5, 2]$, resulting in $s_i^{l+1}$ values for groups ranging from $i = 0$ to $i = 3$ being $[0, 1, 1, 1]$.

Thus, at layer $l$, the $g_l$ groups of channels can directly compute the feature maps in a single step. If the number of groups between two adjacent layers $l$ and $l+1$ is different, then the last output in layer $l$ needs to generate $g_{l+1}$ parallel spike trains.

The workflow of a layer incorporating the PSG algorithm is detailed in Algorithm 1.

---

**Algorithm 1** A Layer with Parallel Spike Generation Model

---

**Require:** The output groups of spike trains $s_i^{l-1}$ for $i \in \{1, 2, \ldots, g_l\}$ from the $(l-1)^{th}$ layer;
  The spike threshold $\theta, \theta \in \{2^n \mid n \in \mathbb{N}\}$;
  The weight $W_l$ and bias $b_l$ for layer $l$.
**Step 1:** Get the average output of layer $l$.
$x^l = \sum_{i=1}^{g_l}(W_l s_i^{l-1}) + b_l)$
$x_{avg}^l = x^l / g_l$ {$g_l$ can be merged into $W_l$ to avoid division operations}
**Step 2:** Calculate $g_{l+1}$ membrane potentials:
$V_i^l = ((i-1)x_{avg}^l \bmod \theta) + x_{avg}^l, i \in \{1, 2, \ldots, g_{l+1}\}, \theta \in \{2^n \mid n \in \mathbb{N}\}$
**Step 3:** Generate $g_{l+1}$ spike trains in parallel:
$s_i^l = \begin{cases} 1, & \text{if } V_i^l \geq \theta \\ 0, & \text{otherwise} \end{cases}, i \in \{1, 2, \ldots, g_{l+1}\}$
**return** $s_i^l$ for $i \in \{1, 2, \ldots, g_{l+1}\}$ - input spikes of layer $l+1$.

---

**Performance Advantage.** By leveraging the average of the output from the previous layer, our PSG model is more accurate for preserving the information and more robust than the IF model when the time window size reduces. As an illustration, consider an example with an element in the output $x_{avg}^l = 1.5, g_{l+1} = 4, \theta = 2$, the spike train generated by our method is $s_{PSG}^l = [0, 1, 1, 1]$, representing 1.5 accurately with three out of four spikes and a threshold $\theta = 2$. However, considering when the output before averaging as $x^l = [0.5, 1, 2, 2.5]$ whose average is also 1.5, the spike train generated by the IF model would be $s_{IF}^l = [0, 0, 1, 1]$, which represents 1. $s_{IF}^l$ loses information when generating spikes, especially at the end of the time window when remaining potentials are dropped. The averaging operation is also used by (Yan et al., 2022) for SNNs with $T > 1$. However, taking an average may be incompatible with the operations of SNNs. OneSpike solves this problem and the resulting SNNs fully conform to all the working principles of SNNs.

## 5 EXPERIMENTS

### 5.1 EFFECT OF PARALLEL SPIKE GENERATION

We first contrast the performance between the PSG method and the IF model. For a spike generation method, its objective is to closely approximate the scale value output by the corresponding ANN. We conduct tests using the CIFAR-10 dataset on the VGG-11 and VGG-16 models(Simonyan & Zisserman, 2014). We trained the VGG models and converted them into SNNs with the IF model using the ANN-to-SNN conversion method mentioned in 3.2. Subsequently, we further covert this

model into SNNs with the PSG method(e.g. a OneSpike model), ensuring that both models share identical weights and structures.

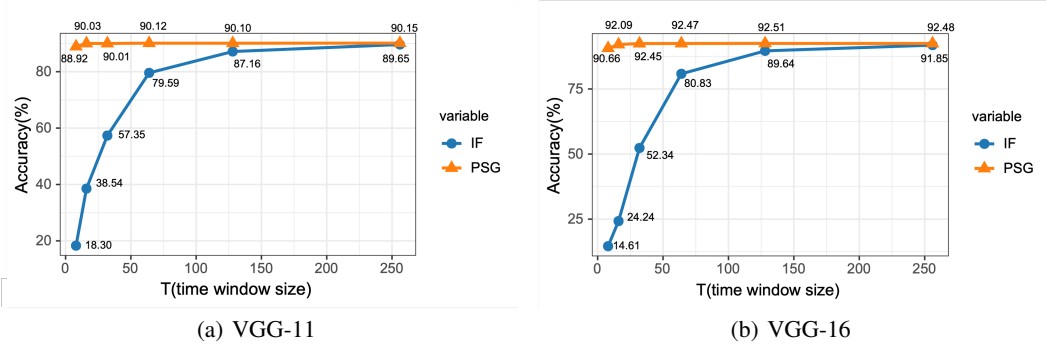

(a) VGG-11  (b) VGG-16

Figure 2: PSG and IF comparison. Timesteps are 8, 16, 32, 64, 128 and 256

Figure 2 displays the accuracies of both SNNs. The graph reveals that at larger $T$, specifically 128 and 256, both the IF model and PSG demonstrate comparable performances. However, as the timestep diminishes, the IF model struggles to accomplish the task, while the PSG consistently preserves a superior accuracy rate. For low-latency SNNs, the PSG method offers a performance advantage over the IF model.

## 5.2 EXPERIMENT SETUP

We evaluate OneSpike on the ImageNet dataset (Deng et al., 2009), one of the most complex image classification datasets commonly used. OneSpike is compared against baselines from both ANN-to-SNN conversion methods and direct training methods, which validate the effectiveness of OneSpike and our model pruning efforts. To facilitate and standardize our training process, we adopt some of the automated data augmentation methods utilized by RepVGG. Our experiments are performed on NVIDIA A100 GPUs, based on Pytorch(Paszke et al., 2019) version 1.12.1, RepVGG(Ding et al., 2021), and Timm(Wightman, 2019).

For ANN pre-training, we employ the model reparameterization technique from RepConv(Ding et al., 2021), which is currently considered one of the leading training methods for plain convolutional neural networks (CNNs). RepConv enables the combination of multiple computational modules into a single module during inference, thereby enhancing accuracy. In line with RepConv, we utilize a combination of one 3×3 convolution, one 1×1 convolution, and an identity connection within a single convolutional layer to achieve higher accuracy.

## 5.3 MODEL STRUCTURE

To enhance baseline accuracy and simplify training, we utilize RepVGG-L2pse (Ding et al., 2021) along with its number of blocks, and its pre-trained weights obtained using their proposed reparameterized training method. However, we exclude the squeeze and extract blocks from their model as they are incompatible with the SNN architecture.

In the presented OneSpike model, there are five stages, excluding the output average pooling and fully connected layers. The number of 3x3 convolutional layers of each stage is [1, 8, 14, 24, 1]. The architectures of OneSpike models are listed in Table 2.

Table 2: Model Architecture

| Model | Channel number |
|---|---|
| OneSpike-8 | [160, 160, 320, 640, 2560] × 8 |
| OneSpike-16 | [160, 160, 320, 640, 2560] × 16 |
| OneSpike-32 | [160, 160, 320, 640, 2560] × 32 |

Our models do not contain any operations incompatible with SNNs. Batch normalization and division operations are folded into the model weights, and complex activation functions such as swish and softmax are not used.

## 5.4 RESULT

Our experiment results are presented in Table 3. Our OneSpike-32 model achieves a top-1 accuracy of 81.92% in $T = 1$. Using OneSpike-8, which is only one-quarter the size of OneSpike-32, an accuracy of 75.92% can be achieved.

Table 3: OneSpike Evaluation on ImageNet.

| Model | Description | Accuracy | Timestep | FLOPs(Billions) |
|---|---|---|---|---|
| OneSpike-8 | 8 groups, $\theta = 2$ | 75.92 | 1 | 30.1 |
| OneSpike-16(2) | 16 groups, $\theta = 2$ | 80.24 | 1 | 60.2 |
| OneSpike-16(4) | 16 groups, $\theta = 4$ | 78.86 | 1 | 60.2 |
| OneSpike-32 | 32 groups, $\theta = 4$ | 81.92 | 1 | 120.4 |

**Comparison with the state-of-the-art SNNs using ImageNet.** Table 4 compares OneSpike with the state-of-the-art methods on the ImageNet dataset, demonstrating superior performance in terms of both accuracy and latency. Among these methods, ANN-to-SNN has traditionally used large time steps, possibly exceeding 200, resulting in high latency and energy consumption, in exchange for higher accuracy compared to direct training methods. Notably, in the works (Zheng et al., 2021; Datta et al., 2022; Fang et al., 2021; Hu et al., 2021; Bu et al., 2023) that adopt ResNet for SNNs, the accuracy is limited despite using small time window sizes. In particular, most previous works have not considered deep networks. Although (Datta et al., 2022) also achieved single-timestep classification on ImageNet, OneSpike achieved over 10% higher accuracy than their reported results.

Table 4: Comparison with various SNNs on ImageNet. '*' denotes the use of complex attention layers, making counting less straightforward.

| Method | Layers | Accuracy | Timestep | Params (in millions) |
|---|---|---|---|---|
| Direct Training (Zheng et al., 2021) | 34 | 67.05 | 6 | 21.80 |
| Direct Training (Datta et al., 2022) | 50 | 66.32 | 1 | 25.56 |
| Hybrid(Chowdhury et al., 2021) | 16 | 67.71 | 1 | 138.36 |
| Direct Training (Datta et al., 2022) | 16 | 68.00 | 1 | 138.36 |
| Direct Training (Fang et al., 2021) | 152 | 69.26 | 4 | 60.19 |
| ANN-to-SNN(Ho & Chang, 2021) | 16 | 70.75 | 35 | 138.36 |
| ANN-to-SNN (Hu et al., 2021) | 50 | 72.75 | 350 | 25.56 |
| ANN-to-SNN(Bu et al., 2023) | 34 | 73.37 | 256 | 21.80 |
| ANN-to-SNN(Bu et al., 2023) | 16 | 74.22 | 256 | 138.36 |
| Direct Training (Zhou et al., 2022) | * | 74.81 | 4 | 66.34 |
| Direct Training (Zhou et al., 2023) | * | 75.85 | 4 | 66.34 |
| ANN-to-SNN (Li et al., 2021a) | 23 | 77.50 | 256 | 22.1 |
| ANN-to-SNN(Yan et al., 2022) | 28 | 79.16 | 200 | 94.08 |
| OneSpike-32 | 48 | **81.92** | **1** | 118.11 |

**Model parameter and FLOPs.** While our focus is on achieving high accuracy with the smallest time step for SNNs, we also consider the size and computational demand of the model. All OneSpike models have 118 million parameters, and their theoretical Floating Point Operations (FLOPs) values are calculated and listed in Table 3. The spike rate on the ImageNet test set for OneSpike is 11.46%, which is used for our calculations.

The parameter count and FLOPs of some popular models, calculated by a PyTorch toolkit called OpCounter(Lyk), are compared with our model in Table 5. In Table 5, $n$ represents the equivalence of energy consumption between $n$ additions and one multiplication. OneSpike has a parameter num-

ber comparable to common CNN models, yet it exhibits a significant advantage in computational complexity due to all the multiplications being replaced by additions.

Table 5: The number of parameters (in millions) and FLOPs (in billions) of the different models. $n$ is the number of additions considered computationally equivalent to 1 multiplication. The exact ratio depends on many implementation details.

| Model | VGG-16 | ResNet-152 | RepVGG-B3 | OneSpike-8 |
|---|---|---|---|---|
| No. of Parameters | 138.36 | 60.19 | 110.96 | 118.11 |
| FLOPs | $15.5n$ | $11.58n$ | $26.2n$ | |
| FLOPs($n = 3$) (Fog, 2022) | 46.5 | 34.74 | 78.6 | 30.1 |
| FLOPs($n = 4.1$) (Horowitz, 2014) | 63.55 | 47.48 | 107.42 | |

## 6    DISCUSSION

### 6.1    COMPARISON WITH BINARY NEURAL NETWORKS

Given the similarity between the single-time step OneSpike and BNN, as both involve binary activation, we also compare OneSpike with the state-of-the-art BNNs on ImageNet in Table 6. Although in terms of binary weights, BNNs exhibit relatively higher energy efficiency, OneSpike achieves significantly higher accuracy, surpassing BNNs by over 10%.

Table 6: Comparison of OneSpike with BNN methods

| | Top-1 Accuracy (%) |
|---|---|
| Bi-real (Liu et al., 2018) | 62.20 |
| AdaBin (Tu et al., 2022) | 66.40 |
| Real-to-Bin (Martinez et al., 2020) | 65.40 |
| OneSpike | 81.92 |

### 6.2    MODEL SIZE AND WEIGHT QUANTIZATION

To further improve efficiency, we quantized the weights of OnSpike models following the quantization scheme in Yan et al. (2023) from FP32 to INT16. After quantization, OneSpike-8 and OneSpike-16(2) still attained an accuracy of 73.92% and 80.12%, respectively, on ImageNet.

### 6.3    HARDWARE FEASIBILITY.

OneSpike is hardware-friendly in the following ways. Firstly, $\theta$ are constrained to be powers of 2. Consequently, the modulo operation in binary can be efficiently accomplished by reading the least significant bits. In particular, all OneSpike models used in this paper use thresholds that are either 1, 2, or 4. No impact on accuracy was observed. Secondly, we require obtaining $i - 1$ instances of $x_l$, given that $i - 1$ is an integer not exceeding the original window size $T$ of the original SNN model. This requires only at most $\log(T)$ additions for each group.

## 7    CONCLUSION

In this paper, we introduced the parallel spike generation method (PSG) that generates spikes using spike trains in a single timestep for both convolutional layers and fully connected layers. Building upon PSG, we developed the OneSpike framework that transforms SNNs with $T > 1$ into equivalent $T = 1$ SNNs. In addition to achieving the lowest possible latency, OneSpike outperforms the classic IF model due to its ability to better conserve spike information, while retaining all the hardware-friendly features of traditional SNNs. Our OneSpike model attained a top-1 accuracy of 81.92% on the ImageNet dataset, which is the highest reported accuracy achieved on ImageNet for SNNs, as well as significantly outperforming BNNs.

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
