# 1 DETAILED CLAMP/QUANTIZATION LEVEL

Each group number $g$ can be calculated by the clamp level $T_c$ and quantization level $T_q$ (Yan et al., 2021). The clamp and quantization levels used for different groups in the models are listed in Table 1.

Table 1: Channel group and clamp/quantization level mapping table.

| Model | Group Number | Clamp Level | Quantization Level |
|---|---|---|---|
| OneSpike-8 | 8 | 2 | 4 |
| OneSpike-16(2) | 16 | 2 | 8 |
| OneSpike-16(4) | 16 | 4 | 4 |
| OneSpike-32 | 32 | 4 | 8 |

# 2 FLOPS CALCULATION METHOD

For our OneSpike models, we also calculate the model parameters and the FLOPs with the same PyTorch toolkit OpCounter we used for the other baselines. It is also known as THOP (Lyk). We utilized this toolkit to calculate the FLOPs of the ANN model as $F_{\text{ANN}}$ prior to channel duplication. Then the FLOPs for our OneSpike model can be expressed as $g \times r \times F_{\text{ANN}}$, where $g$ is the number of the channel groups and $r$ represents the spiking rate.