# OpenReview forum: "OneSpike: Ultra-low latency spiking neural networks"
_ICLR.cc/2024/Conference — ICLR 2024 Conference Withdrawn Submission_

### Official Review · Reviewer_2hLo · 2023-10-16

**Soundness:** 3 good
**Presentation:** 4 excellent
**Contribution:** 1 poor
**Rating:** 3
**Confidence:** 5

**Summary:**

The paper proposes a method to convert rate-encoded spiking neural network into an equivalent OneSpike model with only one timestep. Authors use a parallel spike generation (PSG) method and develop a OneSpike framework. The paper claims that this method can achieve ultra-low latency, high accuracy, and hardware feasibility for SNNs. The paper compares OneSpike with various state-of-the-art SNNs and BNNs, and shows that OneSpike achieves the highest accuracy (81.92% on ImageNet) over other ANN-SNN conversion methods.

**Strengths:**

Authors evaluation the OneSpike method on ImageNet with RepVGG-L2pse architecture and achieve an 81.92% accuracy.

**Weaknesses:**

Compared to IF neuron, OneSpike neuron use different group to generate spike output corresponding to different timesteps in IF neuron. Thus, the claim of one timestep neuron is not true.

OneSpike model is mathematically equivalent to an activation quantized model. Compare to the widely used, GPU friendly network quantization technique, I don't think OneSpike has any advantage.

**Questions:**

Please discuss the concerns addressed in weakness.

---

> ### Author Response · Authors · 2023-11-21
> **Response**
>
> **Comparison with IF neuron**
>
> Response: Please refer to Section 1.3 above.
>
> **Advantage compared to other quantized networks**
>
> Response: We have conducted comparisons of our models with activation quantized models, such as Binary Neural Networks (BNNs). For further explanation, please refer to Section 1.1 above.

---

### Official Review · Reviewer_t2PL · 2023-10-20

**Soundness:** 1 poor
**Presentation:** 2 fair
**Contribution:** 1 poor
**Rating:** 3
**Confidence:** 5

**Summary:**

This paper proposes a parallel spike generation (PSG) method that generates all spikes for a network layer within a single timestep.

**Strengths:**

1. The authors think that they have achieved superior results on complex datasets under low time latency.

**Weaknesses:**

1. **I think the concept of OneSpike proposed in this paper is actually a gimmick.** As shown in Fig.1, the authors split the same parameters of each layer into $g_l$ groups, in fact speculating whether neurons will fire a spike at $i$-th step ($i=1,...,g_l$) under the condition that the input current in each step is completely the same (i.e. the current is uniformly distributed). Subsequently, they obtain an accurate spike sequence $s_1,...,s_g$ under the condition of uniform input current, then continue to calculate the new average input current $x^{l+1}$ after passing through the next-layer weights $W^l$. Note that in this process $s_1,...,s_g$ are respectively calculated with $W^l$ and the overall number of operations is the same as the number of operations in the previous works that emitted spikes for $g_l$-steps. **That is to say, the overhead of OneSpike mentioned by the authors is actually equivalent to the cost of the previous researchers' $g_l$ time-steps.**

2. Eq.10 involves multiplication and modulus operations, which were usually not allowed in previous SNN related works.

3. The reason why authors can achieve an accuracy >80% is not merely because their algorithm is superior to previous works, but because of the advantages of RepVGG network structure itself. Previous works mainly used VGG-16 and ResNet-34, which is obviously difficult to achieve an accuracy of >75% on ImageNet.

Overall, I think the contribution of this paper is actually very limited. If we switch the order of the weight matrix $W^l$ and summation operations ($\sum$) in Figure 1, in fact, the operation mechanism of the entire network is completely equivalent to QCFS ANN [1], which is an ANN with quantized activation output.

[1] Tong Bu, Wei Fang, Jianhao Ding, PengLin Dai, Zhaofei Yu, and Tiejun Huang. Optimal ann-snn conversion for high-accuracy and ultra-low-latency spiking neural networks. ICLR 2022.

**Questions:**

See Weakness Section.

---

> ### Author Response · Authors · 2023-11-21
> **Response**
>
> **Neurons are speculated to fire spikes uniformly across steps, leading to a spike sequence and average input current calculation with the same operational cost as previous multi-step methods**
>
> Response: Thank you for acknowledging the accuracy of the spike sequence generated by our model. Compared to previous works that operate over multiple $g_l$ timesteps, our model performs all computations in parallel. This has the advantage of a better global picture.
>
> Compared to the traditional IF model, PSG can more accurately approximate the scaled value in the ANNs as shown in Section 5.1 of the main paper.
>
> **The multiplication and modulus operations involved in the Eq.10**
>
> Response: Eq. 10 is the full precision math that needs to be done, not the actual implementation. We used the power of 2 values, simpler shifts, bitwise operations etc. to approximate it. Details are discussed in Section 4.2 of the main paper.
>
>
>
> **The advantages of RepVGG network structure**
>
> Response: We chose baselines with high accuracies because they are state-of-the-art. The key is that our method will convert any multi-step SNN into a single-step one with little loss of accuracy.
>
>
>
> **QCFS ANN**
>
> Response: QCFS primarily focuses on ANN-to-SNN conversion, whereas OneSpike starts with a SNN. Also, QCFS requires a relatively large timestep of 16 to achieve an accuracy of only 50.97% on ImageNet. If we had started the same SNN as QCFS, we expected to achieve a higher accuracy but with a single timestep. Unfortunately, we were unaware of the QCFS work and did not have the time to do that.

---

### Official Review · Reviewer_nUrX · 2023-10-27

**Soundness:** 2 fair
**Presentation:** 2 fair
**Contribution:** 2 fair
**Rating:** 3
**Confidence:** 3

**Summary:**

The paper represents an interesting step in the efforts to make SNNs live up to the promise of lower energy consumption than there ANN counterparts. The paper proposes an ANN-to-SNN conversion, or more specifically a conversion from N-step SNNs to 1-step SNNs that preserves accuracy. The results on ImageNet getting over 80% accuracy is really strong as this is a much higher accuracy than previous SNN papers. However, in this reviewer's perspective there are several questions unanswered which makes the true benefits of the approach unclear.

**Strengths:**

The resulting accuracy improvement on ImageNet is impressive.

**Weaknesses:**

The weaknesses of the paper are related to an incomplete energy model and analysis. It does not fully consider the cost of memory access nor the cost of handling the sparsity (compared to ANNs).  Comparisons to state of the art SNNs are focused on accuracy and not energy.

**Questions:**

The paper talks proposes to in parallel create different spike groups changing the traditional IF model significantly. However, this opens the question of proper comparisons. For example, there are many non-multiplier-based implementations of ANNs that should also be considered when doing comparisons. In particular, the fact that their approach involves a module of a power of 2, made me think that their approach must be similar to decomposing the weights of an ANN bit-wise. However, I understand that the power of 2 for each group and each layer is fixed. I had wondered if you considered varying theta for different groups.

More generally, I think it would be good for the paper to better explain the SNN -> ANN conversion step. Your abstraction mentions this but in your algorithm, you focus on converting a N-step SNN to a 1-step SNN. Also, in figure 1, it seems you are using W for both weights and a dimension of the input feature map. This is confusing. Can you clarify?

I find the analysis of energy consumption based on FLOPs somewhat limiting. In many neuromophic designs the dominant energy consumption is the weight and membrane potential lookup. Can you include an estimate of the memory access cost in your designs and comparisons? There are a number of energy models of SNNs (see e.g., https://arxiv.org/pdf/2309.03388.pdf) that include means of capturing the memory cost of SNNs that would make the results far more reliable.  In particular, my concern is that most of the membrane potentials need to be updated despite the sparsity of activations and this should be captured.

Secondly, I think the paper should at least have a discussion of  the cost of supporting the SNN bit-level sparsity (compared to ANNs that do not do typically have or need to handle this granularity of sparsity).  For example, looking up a 1-bit activation is not 8 times less energy than looking up a 8-bit activation because much of the energy is associated with address decoding.  In designs that are spike centric (like Loihi) the cost of memory lookups and routing data can overshadow the cost of add vs mulitply (which is why they support graded spikes). Numerous hardware designs have been proposed to better manage weight and activation sparsity but they come at a cost. This should be recognized when proposing advanced SNN algorithms.

I also wondered if your constraint on the ANN quantization has an impact on accuracy. This does not seem to be addressed.  It was called "near-lossless" but not quantified (from what I can see).  Can you clarify?

---

> ### Author Response · Authors · 2023-11-21
> **Response**
>
> **ANN comparisons and the possibility of varying theta across groups.**
>
> Response:
>
> + We have compared our model with BNNs and listed the advantages of the OneSpike model in the Discussion section. For further explanation, please refer to Section 1.1 above.
>
> + The modulo operation is needed to ascertain the residual membrane potential for each group. Specifically, this involves taking the modulo of the accumulated membrane potential with the threshold to simulate the firing process post-spike. However, the direct use of it is not compatible with SNNs. Therefore, in our design, all thresholds for OneSpike are powers of two. Modulo 2 is then simply examining the least significant bit of a binary number. However, it is not for decomposing the weights of ANNs, as the reviewer asserted.
>
> + We did not apply different theta values for each group because they share the same weight and input. We use different groups to represent the same batch of data more precisely compared to the IF model. The original purpose of using a fixed theta for each group is to make the output more consistent, however, using different theta among layers could be a potential way to improve the energy efficiency and the flexibility of our model.
>
>
>
> **The explanation of ANN-to-SNN conversion and the update of Figure 1**
>
> Response: Please refer to our opening response. OneSpike will work with any SNN, regardless of how it was obtained.
> Thank you for pointing out our mistake in Figure 1, we already updated our figure.
>
>
>
> **FLOPs based energy evaluation**
>
> Response: We agree with the reviewer and have responded to the issue of energy in the last paragraph of the general response. There are many aspects of hardware that, short of actually building it, are rather hard to quantify. Furthermore, a detailed discussion of these would be outside the scope of ICLR. Hence, our use of FLOPS as an approximation.
>
> **Constraint on the ANN quantization having an impact on accuracy**
>
> Response: Thank you for pointing this out. As our focus is on SNN-to-SNN conversion, the specifics of ANN-to-SNN conversion do not constitute our primary concern. Any method that develops a multi-step SNN can be adopted. In Section 3.2, we introduced the method for ANN-to-SNN conversion used in our experiments. This method includes processes of clamping and quantization. For details, please refer to the following paper:
>
> Z. Yan, J. Zhou and W. -F. Wong, "CQ$^{+}$ Training: Minimizing Accuracy Loss in Conversion From Convolutional Neural Networks to Spiking Neural Networks," in IEEE Transactions on Pattern Analysis and Machine Intelligence, vol. 45, no. 10, pp. 11600-11611, Oct. 2023, doi: 10.1109/TPAMI.2023.3286121.

---

> > ### Comment · Reviewer_nUrX · 2023-11-22
> > **Thank you for your response.**
> >
> > Thank you for your rebuttal. I appreciate the clarification of that the approach can handle any SNNs and the correction to Figure 1. I think my biggest concern is that the benefit of the approach remains unclear - while you do achieve one time step, you incur additional costs that I think cannot be reflected in FLOPs alone.  While I agree that reducing the computation to 1 time step helps,  the addition of groups may make data movement and memory lookups more complex.  it is hard to estimate many of these aspects, several papers have proposed unified energy models that at least take into account memory lookups (relating the energy of a memory lookup to that of something like an 8-bit add). I think other reviewers have similar concerns.

---

> > > ### Author Response · Authors · 2023-11-23
> > > **Thank you for your timely reply**
> > >
> > > Thank you for your timely reply. We agree that the use of FLOPS is at best a simple approximation. But as the reviewer pointed out, a detailed energy model is very involved and would constitute another paper altogether. However, perhaps we forgot to mention that we had in mind a weight-stationary processor which makes more sense since activations are sparse and hence there is less of them to move around, although admittedly routing them may become an issue. Still, the emphasis of the current submission is the model itself and the machine learning level KPIs that it is capable of achieving.

---

### Official Review · Reviewer_yNko · 2023-10-30

**Soundness:** 3 good
**Presentation:** 3 good
**Contribution:** 3 good
**Rating:** 6
**Confidence:** 3

**Summary:**

The authors propose a method to convert a classical analog neural network into a spiking neural network. The results are compared with other state-of-the-art methods and show good performance on the imagenet challenge.

**Strengths:**

The paper is clear and well written. The results are interesting and show a significant performance improvement over state-of-the-art methods.

**Weaknesses:**

The parallel spiking generation method could be understood as some kind of quantization of analog numbers in a dyadic format, and this point should be more clearly stated.  As such, such a method seems relatively similar to methods that use quantization in analog networks. In particular, even if this method seems original, the parallelism with existing methods needs to be strengthened. In particular the claim that "To the best of our knowledge, this study is the first to explore converting multi-timestep SNNs into equivalent single-timestep ones" should be circonstantied. On the other hand, do you see any analogy between this mechanism and processes that might take place in biological neural networks? It seems, for example, that predictive methods will use residuals, and that these can themselves be quantified, and so on... but to my knowledge, there are no papers exploring this interesting direction of research.

**Questions:**

The method presented in this paper works well on static images. Do you think this method could be extended to dynamic images, such as videos? Do you think this method could be extended to recurrent neural networks?


Minor:
- "a PyTorch toolkit called OpCounter(Lyk), " fix reference

---

> ### Author Response · Authors · 2023-11-21
> **Response**
>
> **Difference between PSG and quantized value in analog network and the parallelism with existing methods.**
>
> Response: All SNNs can be seen as quantization. What makes PSG stand out is its consideration of temporal dynamics.
> The parallel spiking generation method resembles the quantization of analog signals, particularly in the way it discretizes continuous information. Unlike SNNs, quantized neural networks (particularly binary networks) typically do not consider temporal dynamics. For the comparison with quantized models and the IF model, please refer to Sections 1.1 and 1.3 of the general response above.
>
> **The claim of the first to explore converting multi-timestep SNNs into equivalent single-timestep ones.**
>
> Response: With all due respect, we stand by this claim as so far we have not found any evidence to the contrary.
>
>
> **Analogies with processes in biological neural networks and extensions to dynamic images and recurrent neural networks.**
>
> Response: Thank you for pointing out the potential of exploring the analogies between our PSG method and biological neural processes as well as extending to dynamic images or videos, this may be our future direction.

---

### Author Response · Authors · 2023-11-21
**General Response**

We would like to thank the reviewers for their comments, some of which are invaluable to improving the paper. We shall first respond to common queries that more than one reviewer had asked, before responding to the individual reviewers.

Our work converts any multi-timestep SNN to an equivalent one that uses only a single timestep, with just a little worse, and at times better, accuracy. In doing so, we preserve the hardware friendliness of SNNs - that the method continues to be multiplication and division free, and does not use any complex activation function. It is important to note that our workflow starts with an SNN. That initial SNN can be derived by direct training, proxy training, or conversion. We so happen to use one of the state-of-the-art ANN to SNN conversion methods to get the initial SNN. The effectiveness of the ANN-to-SNN conversion, or issues regarding SNNs, is not our work's focus. More than one reviewer highlighted the issue regarding the distinction between OnesSpike and quantized neural networks like BNNs, as well as the feasibility of implementation on hardware. These are indeed relevant to our contribution and design, and to which we will respond here.

## 1.1 Compared to the quantized models
Compared to GPU-optimized quantized models such as binary neural networks (BNNs), OneSpike has two key differences: Firstly, our OneSpike models significantly outperform state-of-the-art BNNs in terms of accuracy. The state-of-the-art BNNs achieved around 65\% accuracy on the ImageNet dataset, while OneSpike achieved 81\% accuracy. We attribute this to OneSpike's PSG method which preserves the temporal sequence characteristics inherent to SNNs across different groups, enabling better retention of neural representations. Secondly, unlike quantized models that are typically meant for GPUs, SNNs are inherently more compatible with neuromorphic hardware, or custom ASIC hardware. OneSpike goes beyond simple activation quantization by also doing away with the use of complex computations like swish and softmax. Consequently, our approach yields advantages in energy efficiency and processing speed in the form of reduced computational load, especially for neuromorphic hardware, even with equivalent model sizes, or more efficient custom hardware.


## 1.2 Hardware efficiency
OneSpike is (custom) hardware friendly. We adhered strictly to the design principles of SNNs. In our process of computing the membrane potential for each group, we use a power of 2 value as the threshold. This simplification facilitates the computation of the remaining membrane potential, allowing the spike and subtraction operation to be efficiently implemented by shifting.
We thank Reviewer nUrX for pointing out that in the reference "Are SNNs Truly Energy-efficient? — A Hardware Perspective," which provides insightful perspectives on the energy consumption of SNN models, the article indeed mentions several challenges beyond FLOPs that may contribute to computational overhead and increased energy consumption in SNNs. These include increased timesteps on SATA hardware leading to additional computation and data movement costs, the hardware expense of generating output spikes and storing membrane potential across timesteps on SpikeSim, and the potential for error accumulation in dot-product operations over multiple timesteps. Furthermore, the inference energy and latency on SpikeSim are noted to scale linearly with timesteps. These challenges identified by the paper precisely reinforce the value proposition of OneSpike - converting any multistep SNNs into a single step one.

## 1.3 Compared to the IF model
Compared to the existing integrated-and-fire (IF) model for SNNs, our approach ensures that *all* outputs of a single layer are generated in a single timestep in parallel. The traditional IF model sequentially processes each timestep. Other than what's remaining in the membrane potential, it has no memory of spikes that had happened. Secondly, by allowing global information to be processed concurrently, the spike train generated by our method is more accurate than that of IF neurons. For instance, consider an input sequence [0.1, 0.1, 0.5, 1.9, 4.9] with a threshold of 2. Setting 4 as the timestep in the IF model yields an output of [0, 0, 0, 1, 1]. In contrast, in the PSG model, the average value calculated is 1.5. This results in the generation of spikes [0, 1, 1, 1, 0], which more accurately reflect the input values and avoid losing any information at the end of the spike train.